# Results of Adding Sludge Micropowder for Microbial Structure and Partial Nitrification and Denitrification in a Filamentous AGS-SBR Using High-Ammonia Wastewater

**Jun Liu [1,2], Dong Xu [3], Weiqiang He [1], Qiulai He [4], Wenhai Chu [5], Songbo Li [1,*] and Jun Li [6,*]**

1   School of Modern Agriculture & Urban Construction, Jiaxing Vocational & Technical College, Jiaxing 314036, China
2   Department of Civil Engineering, Tongji Zhejiang College, Jiaxing 314051, China
3   College of Geomatics & Municipal Engineering, Zhejiang University of Water Resources & Electric Power, Hangzhou 310018, China
4   Department of Water Engineering and Science, College of Civil Engineering, Hunan University, Changsha 410082, China
5   State Key Laboratory of Pollution Control and Resources Reuse, National Centre for International Research of Sustainable Urban Water System, College of Environmental Science and Engineering, Tongji University, Shanghai 200092, China
6   College of Environment, Zhejiang University of Technology, Hangzhou 310014, China
*   Correspondence: lly202204@sina.com (S.L.); tanweilijun@zjut.edu.cn (J.L.)

**Abstract:** This work investigated the roles of sludge micropowder addition in microbial structure and partial nitrification and denitrification (PND) in an extended filamentous aerobic granular sludge-sequencing batch reactor (AGS-SBR) using high-ammonia wastewater. Type 1683 *Acinetobacter* with a high percentage became the dominant extended filaments, remarkably shifted and remained at a low level, acting as a framework for AGS recovery after micropowder addition. The sludge volume index ($SVI_5$) decreased from 114 to 41.7 mL/g, mixed liquid suspended solids (MLSS) and extracellular polymers (EPS) both increased and balanced at 6836 mg/L and 113.4 mg/g•MLVSS, respectively. COD and $NH_4^+$-N were degraded to certain degrees in the end. However, the effluent $NO_2^-$-N accumulated to the peak value of 97.6 mg/L on day 100 (aeration stage), then decreased and remained at 45.3 mg/L with development of the stirring and micropowder supplemented in the SBR on day 160 (anoxic stage), while the influent $NO_2^-$-N always remained at zero. Interestingly, the influent/effluent $NO_3^-$-N both remained at zero throughout the whole experiment. These results demonstrated that PND was successfully obtained in this work. Sludge micropowder addition not only restrained the extended filaments' overgrowth, but also contributed to PND realization with carbon released. *Citrobacter* and *Thauera* played an essential role in the PND process for high-ammonia wastewater treatment. Running condition, wastewater characteristic, and sludge structure played an important role in microbial composition.

**Keywords:** aerobic granular sludge; extended filaments; Type 1683 *Acinetobacter*; sludge micropowder; partial nitrification and denitrification

## 1. Introduction

As a promising biotechnology, aerobic granular sludge (AGS) was formed through microbial self-immobilization without carriers in different bioreactors using various wastewaters [1–4]. AGS is extensively used from lab- to full-scale treatment plants due to the unique characteristics of excellent settleability, compact structure, high biomass content and retention, and resource recovery [5–7]. Sludge bulking reduced AGS structural stability, and threatened granular sludge operation and organic degradation. It was

reported that excessive growth of filamentous bacteria extends out of aerobic granules, causing sludge bulking [8–10].

At present, the overgrowth of filamentous microbes is the chief factor for sludge disintegration in both AGS and conventional activated sludge (CAS) [8,11–13]. The low concentrations of filamentous bacteria in sludge were attributed to organic removal in wastewater treatment plants (WWTPs) [14,15], and served as the backbone to rapidly develop aerobic granules [5,16]. Subsequently, filamentous bacteria played negative and positive roles in sludge structure for both AGS and CAS. Effective actions must be taken to prevent the negative occurring and make good use of the positive for AGS long-term running. These strategies of chemical addition (NaClO, Cl$_2$, or H$_2$O$_2$), ballasting agents, and coagulants' addition have been employed to prevent/control overgrowth of filamentous organisms [13,17–19]. However, these approaches only provide a temporary solution to sludge bulking, and chemical addition not only causes toxic byproducts, but also would kill other bacteria with the functions of sludge aggregation or organic degradation [8,13,20].

Other approaches such as the use of particulate substrates, improving shear force, organic loading rate, and salinity were attempted to mitigate filamentous bacterial growth [21–24]. The addition of particulate substrates played a key role in controlling microbial metabolism and, thus, enhanced sludge settleability [23,25,26]. Dried sludge micropowder as an eco-friendly alternative, and the sustainable particulate matter, was used in an attempt to control the growth of extended filaments in a lab-scale AGS-SBR [27]. The following work revealed that adding dried sludge micropowder could restrain the filaments of Type 021N [10]. Additionally, aerobic granulation was achieved in a continuous-flow reactor treating real and low-strength wastewater by adding sludge micropowder [28]. In brief, these works proved that the filaments were sensitive to sludge micropowder addition and could promote AGS formation by acting as nuclei to induce bacterial attachment.

However, the dynamics of filamentous bacteria were not clear before and after sludge micropowder addition, although it was found that Type 021N took the main responsibility for AGS break-up using filamentous morphology [27,28]. At present, more than 30 different filamentous bacterial strains have been identified [13,29]. As we know, *Thiothrix eikelboomii* and *defluvii* species as the gamma-subclass of *Proteobacteria* belong to Type 021N [13], while *Acinetobacter*, *Anaerolinea*, *Flavobacterium*, *Sphaerotilus*, and *Singulisphaera* also contribute to the filamentous groups [9]. However, earlier works did not tell us the concreated filamentous bacteria [10]. Meanwhile, the transformations of NH$_4^+$-N, NO$_3$-N, and NO$_2$-N during the addition of dried sludge micropowder were not elaborated on in earlier works [27,28] .

It was reported that the wastewater characteristics (excess nutrient/substrate in feed, feed rate, and C/N ratio) played a key role in the overgrowth of filamentous strains [8,9,12,13,29]. The remarkable feature of high-ammonia nitrogen wastewater was a low C/N ratio. Ammonia removal technologies/processes mainly include microalgae-bacteria, activated sludge, integrated fixed-film activated sludge, moving-bed membrane bio-reactors, adsorption, and membrane separation [30]. Additionally, the popular treatment process of high-ammonia nitrogen wastewater is nitrification and denitrification through biological technology. Because of the low C/N ratio, realizing denitrification requires more organic carbon sources, which will increase the treatment cost in practice. The sludge micropowder was added into freshwater and dissolved out to release carbon sources [10,27]. Meanwhile, AGS technology with a 3D structure could create anaerobic/anoxic conditions for organic matter and nutrient (C, N and P) removal under aerobic conditions [31]. Therefore, it is necessary to investigate the role of adding sludge micropowder into a filamentous AGS-SBR with high-ammonia wastewater.

Based on the above analysis, the primary objective of this study was to comprehensively analyze the dynamics of filamentous groups using 16S rRNA technology sludge micropowder addition into an extended filamentous AGS-SBR using high-ammonia

wastewater, as well as identify the influences of sludge micropowder on organic matters' variations of COD, $NH_4^+$-N, $NO_3$-N, and $NO_2$-N in a whole experiment.

## 2. Materials and Methods

### 2.1. Characteristics of Sludge Micropowder

Sludge micropowder with a size of 20–200 μm was made from the municipal wastewater sludge of a wastewater treatment plant in Haining City, China. The sludge micropowder characteristics were determined by dispersing 1 g of dried sludge micropowder in 1 L of fresh water for 2–4 h. The resulting sludge micropowder suspension had a COD value of 60 ± 10 mg/L, in which the soluble and particulate organic matters were 45% and 55%, respectively. The details of the manufacturing operation and characteristics are both presented in earlier works [10,27].

### 2.2. SBR Operation

A lab-scale SBR made of plexiglass with an 11 L working volume (H × D = 50 cm × 20 cm) was used to investigate the roles of sludge micropowder in microbial structure and partial denitrification in this experiment (0–160 days). SBR operation was performed for 6 cycles per day with each cycle lasting 4 h . The details are presented in Table 1. The volumetric exchange ratio was 7/11, the gas velocity was 1.2 cm/s during the aeration period, and the stirring rate remained at 60 rad/min for the anaerobic process. The whole experiment included four stages across 160 days. In the first stage—'A', (days 0 to 60)—filamentous AGS was conducted after the seeding process. In the second stage—'B', (days 61 to 100)—dried sludge micropowder (0.2 g/L of mixed liquid) was added after the discharge on every third day to control the extended filamentous overgrowth. In the third stage—'C', (days 100 to 120)—the stirring (60 rad/min) was carried out with no micropowder addition after the aeration process. This stage was to test the contribution of anoxic processes to N removal rate. For the last stage—'D', (days 121 to 160)—1.0 g/L of micropowder was added once every four days during the stirring process. This was to investigate the role of adding sludge micropowder in N removal efficiency. Generally, the experiment was divided into four stages with each object in every stage.

**Table 1.** SBR operating parameters.

| SBR Process | Running Stages | | | |
| --- | --- | --- | --- | --- |
| | Stage A (Day 0–60) | Stage B (Day 61–100) | Stage C (Day 101–120) | Stage D, (Day 121–160) |
| Influent | | | 5 min | |
| Aeration | | 180 min | | 135 min |
| Stirring (60 rad/min) | | 0 min | | 60 min |
| Settling (min) | | | 5 min | |
| Effluent (min) | | | 5 min | |
| Idling (min) | | 45 min | | 30 min |

### 2.3. Wastewater and Seeding Sludge

The influent was domestic wastewater collected from a septic tank located in a residential community in Hangzhou City, China. The characteristics of the influent were: COD 218–270 mg/L, $NH_4^+$-N 220–290 mg/L, $NO_3$-N and $NO_2$-N 0 mg/L, suspended solids (SS) 200–400 mg/L, pH 7.5 ± 0.5. The wastewater was mainly from residential excrements, e.g., urine, feces and other sewage. This led to the high concentration of $NH_4^+$-N in hot weather with temperatures of 35–40 °C. Subsequently, we often smelled the ammoniacal odor from the influent in summer. This wastewater was not diluted with freshwater in this experiment, which was the main difference from earlier work [10]. The inoculated

sludge was obtained from Qige WWTP, Hangzhou City, China. Additionally, the indoor temperature was in the range of 25 ± 5 °C using air-conditioning (Midea Group Co., Ltd., Foshan, China) throughout the whole experiment.

*2.4. Analytical Methods*

The chemical oxygen demand (COD), ammonium ($NH_4^+$-N), nitrate ($NO_3^-$-N), nitrite ($NO_2^-$-N) mixed liquid suspended solids (MLSS) concentration, and SVI value were measured using standard methods [32]. The extracellular polymeric substances (EPS) were extracted using the formaldehyde-NaOH method [33]. Polysaccharide (PS) concentration in EPS was determined through a phenol-sulfuric acid method using glucose as the standard [34] (Dubois et al., 2002). The protein (PN) concentration in EPS was determined with Coomassie brilliant blue using bovine serum albumin (BSA) as the standard [35].

*2.5. Microbial Sampling and Analysis*

Sludge samples were regularly collected on day 0, 60, 80, 115, and 150, namely S0, S1, S2, S3, and S4, respectively. Each sample was random taken three times and then mixed into one tube (20 mL). All tubes were stored in a refrigerator at −20 °C for microbial analysis. DNA extraction was performed using an sE.Z.N.ATM Mag-Bind Soil DNA Kit (OMEGA, USA) according to the manufacture's instruction. Microbial analysis was conducted at Sangon Biotech (Shanghai Co., Ltd., Shanghai, China) using Illumina Miseq 2 × 300 (PE300, Santiago City CA, USA). About 10 ng DNA of the sludge sample was used for 16S rRNA sequencing targeting the hypervariable V3–V4 regions. The data were processed by the Quantitative Insights into Microbial Ecology (Qiime) pipeline (http://qiime.org/ 20161018) and Mothur (http://www.mothur.org/ 20161018) in this work. Bacterial analysis was carried out based on high-quality sequences with identity more than the 97% threshold. The details were reported in previous work [36].

## 3. Results and Discussion

*3.1. Sludge Characteristics in the Different Stages*

Figure 1 shows sludge sharp's variations with SBR operation in different stages. The seeding sludge sharp was irregular, with a mean size of 20–160 μm (Figure 1a). After 10 days' running, the floccular sludge was washed out the SBR, and a few zoogloeal presented (Figure 1b). Due to the wastewater with low carbon source, microbes grew slowly, and the extended filamentous granular sludge was successfully maintained till day 40 (Figure 1c). After 10 days' development, the filamentous bacteria overgrew out of the surface of the aerobic granules and coexisted with many epistylis (Figure 1d). The domestic wastewater contained particular substances and SS, and epistylis tended to live in this situation, which was in good agreement with an earlier report [37]. In order to restrain filamentous overgrowth, sludge micropowder (0.2 g/L of mixed liquid) was added into the SBR every three days on days 61–100. Namely, 2.2 g of sludge micropowder was supplied into the SBR for extended filamentous control every three days. Then, the extended filamentous bacteria gradually faded away on day 80 (Figure 1f). This result was proven in earlier reports on adding sludge micropowder for filamentous control [10,27]. Therefore, we stopped adding sludge micropowder on day 100. Following this, 1 hour's stirring (60 rad/min) was carried out after the aeration process from day 101 to 120. We found that low-speed stirring did not destroy AGS structure, and the filaments were not observed (Figure 1g). This means that the stirring process was beneficial to the filamentous suppression in this work. On days 121–160, 1g/L sludge micropowder was added into the SBR again every four days for investigation of $NO_2^-$-N removal. Aerobic granules still maintaining integrated sharp and extended filaments were not found during these days (Figure 1g,h).

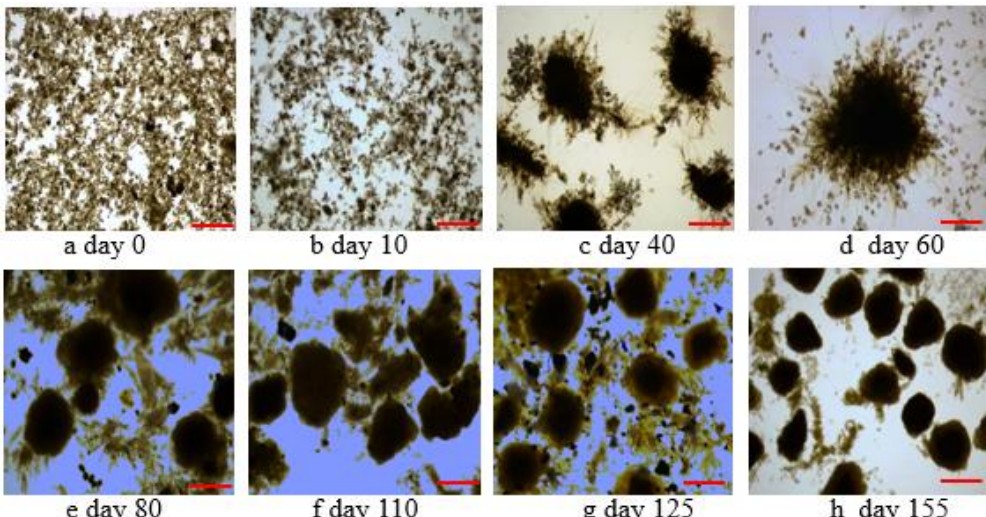

**Figure 1.** Photographs of sludge in different phases, scale bar: 1 cm.

Correspondingly, sludge characteristics including MLSS, SVI, and EPS were also investigated throughout the whole experiment (Figures 2 and 3). As shown in Figure 2, MLSS and SVI have similar tendencies from stage A to D. After the seeding process, MLSS decreased from 3566 mg/L (day 1) to 2447 mg/L (day 20) due to wastewater and operation mode changes. When the microbes adapted to the new environment and grew slowly, MLSS increased to 3178 mg/L on day 30. Additionally, the extended filaments coexisted with AGS on day 40, then MLSS decreased further and to the lowest value, 2088 mg/L on day 60. Meanwhile, $SVI_{30}$ increased from 59.8 mL/g to 114 mL/g on days 1–60, while the $SVI_{30}/SVI_5$ value had a contrary tendency and dropped down from 0.807 to 0.63. This means that some sludge with low settling ability washed out of the SBR and took major responsibility for MLSS reduction on days 1–60. In order to achieve long-term and stable running, 0.2 g/L sludge micropowder was added into the SBR every three days from day 61 to 100. Because of the filaments' sensitivity to sludge micropowder, they were effectively suppressed on day 80 (Figure 1e). Then, $SVI_{30}$ clearly declined and reached 54.3 mL/g on day 100, and the $SVI_{30}/SVI_5$ value increased to 0.91, while sludge remained in the SBR, and MLSS and MLVSS/MLSS increased to 4612 mg/L and 0.75 on day 100. These results were similar to earlier work [10], and further verified that sludge micropowder addition did effectively control the extended filaments of AGS. After that, AGS recovered and developed well, MLSS and MLVSS both kept rising and stabilised at 6836 mg/L and 4990 mg/L on day 160. Correspondingly, $SVI_5$ and $SVI_{30}$ kept decreasing and reached 46.3 mL/g and 41.7 mL/g on day 160 (Figure 2). Interestingly, the dosage of sludge has an extended influence on the MLVSS/MLSS ratio or $SVI_{30}/SVI_5$ ratio between days 61–100 (0.2 g/L every three days) and days 121–160 (1.0 g/L every four days). This is because the sludge micropowder contained organic and inorganic substances for extended filamentous restraining and AGS recovered normally [10,27].

EPS secreted by microorganisms and played a key role in AGS structure. The concentrations of EPS in terms of PN and PS were investigated throughout the whole experiment (Figure 3). Generally, EPS content increased from 59.1 to 79.5 mg/g MLVSS from day 1 to day 30, then decreased to 61.6 mg/g MLVSS on day 60 with the extended filaments existing. When the filaments were controlled by adding sludge micropowder on day 61, EPS steadily increased again and finally reached 113.4 mg/g MLVSS on day 160. Meanwhile, PN and PS had analogical variations, but PS content was still higher than PN content. In particular, PS content was 1.7 times as much as PN on day 100–160. As we know, PN can provide 3D structure, and PS is sticky and acted as a biogel for bacterial adherence [38,39]. Subsequently, AGS with integrated and compact structure finally recovered on day 160 (Figure 1h).

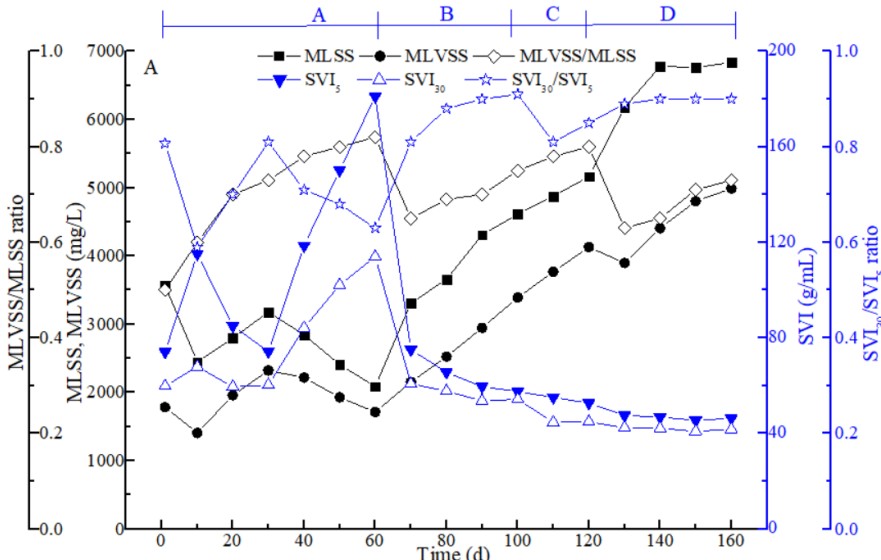

**Figure 2.** Variations of MLSS and SVI during the whole experiment. Stage A—filamentous AGS cultivation, stage B—stable running by adding 0.2 g/L sludge micropowder every 3 days, stage C—adding stirring for anaerobic process at 60 rad/min, stage D—adding 1.0 g/L sludge micropowder for partial denitrification every 4 days.

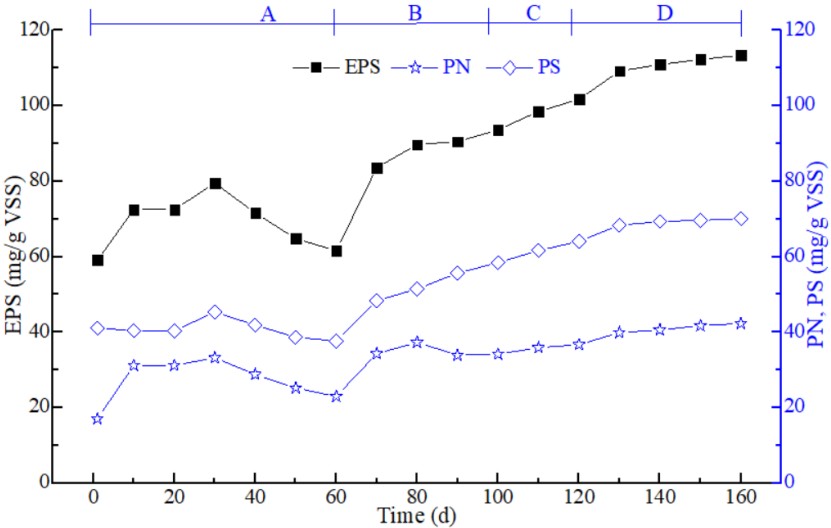

**Figure 3.** Variation of EPS throughout the whole experiment. Stages A–D are the same as in Figure 1.

### 3.2. Removal Performance

Biological treatment performance including COD, $NH_4^+$-N, $NO_3$-N, and $NO_2$-N were all measured throughout the whole experiment (Figure 4). While the wastewater was domestic, influent COD and $NH_4^+$-N had few fluctuations from the ranges of 218–267 mg/L and 221–286 mg/L. Additionally, the influent $NO_3$-N and $NO_2$-N remained at almost zero throughout the whole experiment (Figure 4a). This wastewater was high-$NH_4^+$-N wastewater. The effluent COD and $NH_4^+$-N remained in the range of 109–200 mg/L and 98–172.8 mg/L, respectively. Clearly, CON and $NH_4^+$-N did not completely degrade using AGS processes in this experiment. The COD of domestic wastewater was not very low, but it also contained 45 ± 5% particular organic matters (refractory organics) [10,27]. Meanwhile, the high ammonia might restrain the microbes with key functions of COD and N degradations [9,40,41]. These reasons can be considered for low organic performances on COD and $NH_4^+$-N. Influent and effluent $NO_3$-N almost remained at zero, but the effluent

NO$_2$-N had a rising trend on days 1–100, and reached the peak value of 97.6 mg/L on day 100. Then, it decreased with the stirring process and sludge micropowder addition, finally balancing at 45.3 mg/L on day 160. However, the influent NO$_2$-N always remained at zero throughout the whole experiment (Figure 4a). It was indicated that a portion of NH$_4^+$-N directly transformed into NO$_2^-$N, leading to its accumulation in aerobic processes on days 1–100. This is in good agreement with partial nitrification with high NO$_2^-$-N accumulation [40,41]. The stirring process provided an anoxic environment and sludge micropowder addition supplied a carbon source for enhancing NO$_2^-$-N removal through the biological denitrification process in this work. This result was not reported in earlier works [10,27,28]. This means that adding sludge micropowder is helpful for partial denitrification in this work. As a result of all this, PND was successfully obtained in an extended filamentous AGS system using high-ammonia wastewater after sludge micropowder addition and stirring, although effluent NH$_4^+$-N and NO$_2^-$-N did not meet the discharged standard and need improvement in the future.

On day 150, the AGS-SBR system reached stable reactor performances with an efficient pollutant degradation profile (Figure 4b). The results showed that COD presented decreasing trends in 0–135 min, increased in 136–165 min with sludge micropowder addition and stirring, and finally reduced to 108 mg/L in 195 min. However, NH$_4^+$-N still had a descending trend in the reactor operation. NO$_3^-$-N always remained at zero in whole experiment, but NO$_2^-$-N increased from 0 to the peak value of 65.3 mg/L in 135 min, then decreased and balanced at 45.6 mg/L at 195 min with the supply of sludge micropowder and stirring process (Figure 4b). These results further demonstrated that the PND process was achieved in this work.

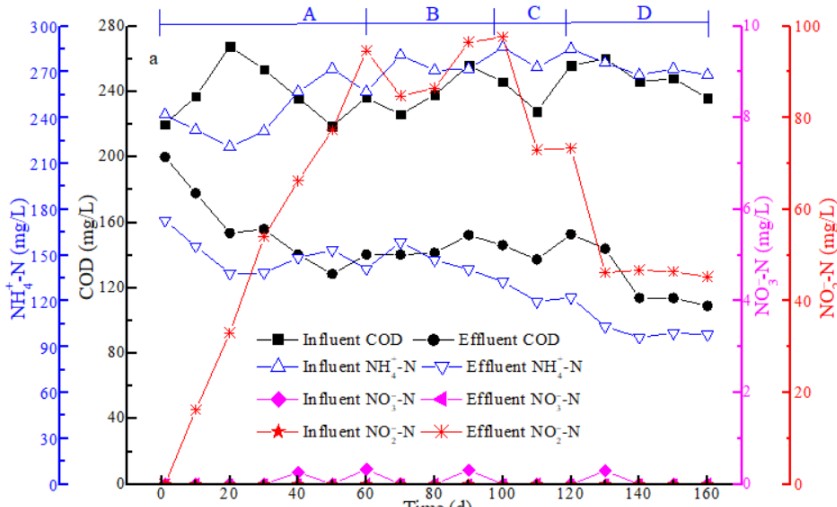

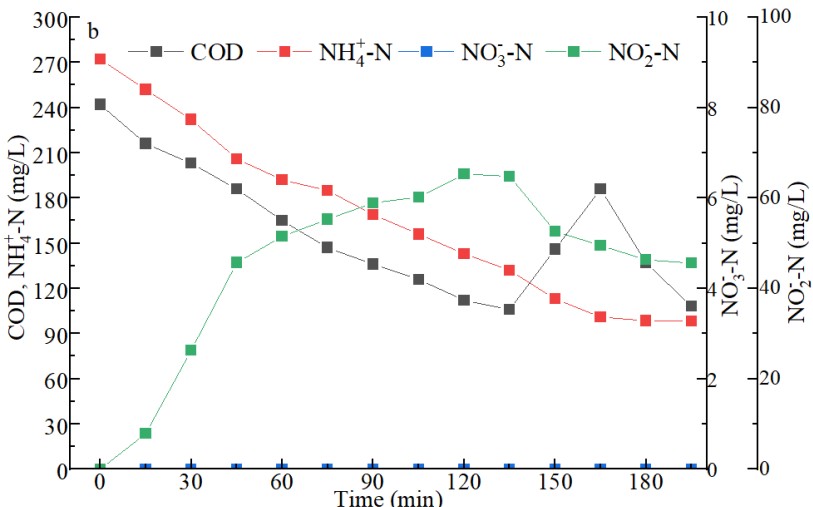

**Figure 4.** Biological treatment performances throughout the whole experiment (**a**), Stages A–D are the same the aforementioned; organic degradation in a typical cycle on day 150 (**b**).

### 3.3. Microbial Diversity and Composition

Five sludge samples (S0–S4) from the SBR were monitored using 16S rRNA technology to explore the dynamics of the microbial community in this work. As shown in Table 2, 42,545–64,649 high-quality reads with a mean length of 415.14–425.5 bp were yielded in all sludge samples, containing a total of 780–1109 operational taxonomic units (OUT) and a sufficiently high depth coverage of 0.99 in our work. Simpson and Shannon imply the microbial diversity, where a lower number of Simpson demonstrates more diversity, while a lower number of Shannon demonstrates less diversity [9,36]. Clearly, sample S2 had the highest Simpson and lowest Shannon values in all the sludge samples (S0–S4), implying that sample S2 had the least microbial diversity. Namely, adding sludge micropowder did effectively restrain the extended filaments and the AGS system recovered on day 80 (S2) in this work. This was in good agreement with earlier works [10,27]. Meanwhile, samples S0, S1, S3 and S4 had similar Shannon and Simpson indices, indicating that the seeding sludge and the normal AGS had similar biodiversity, and adding sludge micropowder might have had significant influence on the bacterial diversity of the extended filamentous AGS. In addition, bacterial richness was also analyzed in terms of ACE and Chao1 indexes (Table 2). Higher ACE or Chao1 represents more bacterial richness. In the present experiment, sample S2 had the lowest ACE and Chao1 indexes, while samples S0–S4 had similar indices. This suggesting the extended filaments were sensitive to sludge micropowder in this work.

**Table 2.** Microbial diversity of all sludge samples (S0–S4).

| Sample | Effective Reads | Mean Length (bp) | OTU | Shannon | Simpson | ACE | Chao1 | Coverage |
|---|---|---|---|---|---|---|---|---|
| S0 | 47,293 | 415.14 | 1109 | 4.6 | 0.03 | 1291.45 | 1247.65 | 0.99 |
| S1 | 42,545 | 422.77 | 780 | 4.35 | 0.03 | 1267.47 | 1104.56 | 0.99 |
| S2 | 47,825 | 425.5 | 784 | 3.3 | 0.17 | 1101.28 | 1063.43 | 0.99 |
| S3 | 55,914 | 417.43 | 1088 | 4.96 | 0.02 | 1292.89 | 1292.90 | 0.99 |
| S4 | 64,649 | 417.53 | 1014 | 4.3 | 0.06 | 1236.15 | 1198.51 | 0.99 |

The phylogenetic classification of the obtained effective reads from five sludge samples (S0–S4) were assigned to various groups at the phylum and genus levels (Figure 5). As shown in Figure 5a, sludge samples S0–S4 included 22, 15, 15, 22, and 22 phyla, respectively. Clearly, *Proteobacteria* were the dominant bacteria in all samples, which was similar to previous works [9,36]. *Proteobacteria* accounted for 30.96% in the seeding sludge (S0), while they increased sharply to 73.85% and 75.35% in the extended filamentous AGS (S1)

and recovered AGS (S3) in our work. This suggested that most of the *Proteobacteria* belonged to the extended filamentous group [13,36], and they were tentatively inhibited by adding sludge micropowder on days 61–100, though this did not mean the filamentous bacteria completely vanished. On days 101–160, stirring and adding more sludge micropowder further restrained the filamentous growth inside the AGS, and the AGS system ran well with compact sharp (Figure 1g,h). Finally, the abundance of *Proteobacteria* balanced at 53.66% (S3) and 55.26% (S4). *Firmicutes* of seeding sludge (S0) possessed the highest relative abundance of 35.87%, but it decreased to 4.72% (S2), then increased to 27.86% (S4). This indicated that wastewater change and adding sludge micropowder limited the filaments of on days 0–100. Additionally, *Firmicutes* belong to the facultative anaerobes, and the stirring process provided the anoxic conditions and AGS recovered with compact structure on days 101–160 (Figure 1f–h). Subsequently, they began to grow and their abundance increased in this work [13]. The rest of the phyla possessed different relative abundances in S0–S4, which might be related to the operation conditions and sludge micropowder addition.

Figure 5b presents the top 50 genera in all samples with their relative abundances in the heatmap. Briefly, there were substantial differences within both the relative abundances and the abundant genera in this work. *Proteiniclasticum* (12.46%) was the most predominant genus in the seeding sludge (S0), then it sharply decreased to 0.04% (S3) and slowly rose to 5.45% (S4). This genus lived in the anaerobic environment, suggesting that exposure to the aerobic was harmful to this microbial growth. Additionally, the stirring process and AGS with compact structure supplying a micro/anoxic-environment were beneficial to its survival. *Methylophilus* (9.08%) was the second most predominant genus in sample S0. It decreased greatly to 0.01% in sample S1, then was not detected in samples S2–S3. As we know, *Methylophilus* live in aerobic environments, and AGS with knit sharp limited oxygen diffusion into the inner area. As a result, *Methylophilus* was not able to survive in this condition and vanished quickly. Additionally, the proportion of *Levilinea* had a clear decreasing trend from 6.08% (S0) to 0.06% (S2), then slowly rose to 1.01% (S4). It was reported that *Levilinea* belonging to *Chloroflexi* exist in anaerobic conditions [13,19]. The operation change and AGS structure created a good living environment for these species in this experiment. This indicated that operation conditions and sludge structure played an important role in microbial composition [42]. Interestingly, *Citrobacter* had significant variation, increasing rapidly from 0.5% (S0) to 39.48% (S3), then decreasing quickly to 0.13% (S4). Meanwhile, the effluent $NO_2^-$-N concentration reached a peak of 97.6 mg/L on day 100, then decreased to 45.3 mg/L on day 160, coupling with the stirring process and addition of sludge micropowder in this work (Figure 4). This is consistent with the previous reports that *Citrobacter* is known as the incomplete denitrifying bacteria and its abundance was close to $NO_2^-$-N concentration [40,43]. As the denitrifiers, *Thauera* had a modest growth in sludge samples S0–S4 (0–3.01%) and became one of the predominant genera in the work (Figure 5b). This was in agreement with the results in earlier works, which suggested that high $NO_2^-$-N in the partial denitrification system might be related to the dominant *Thauera* genus in microbial composition [41,44].

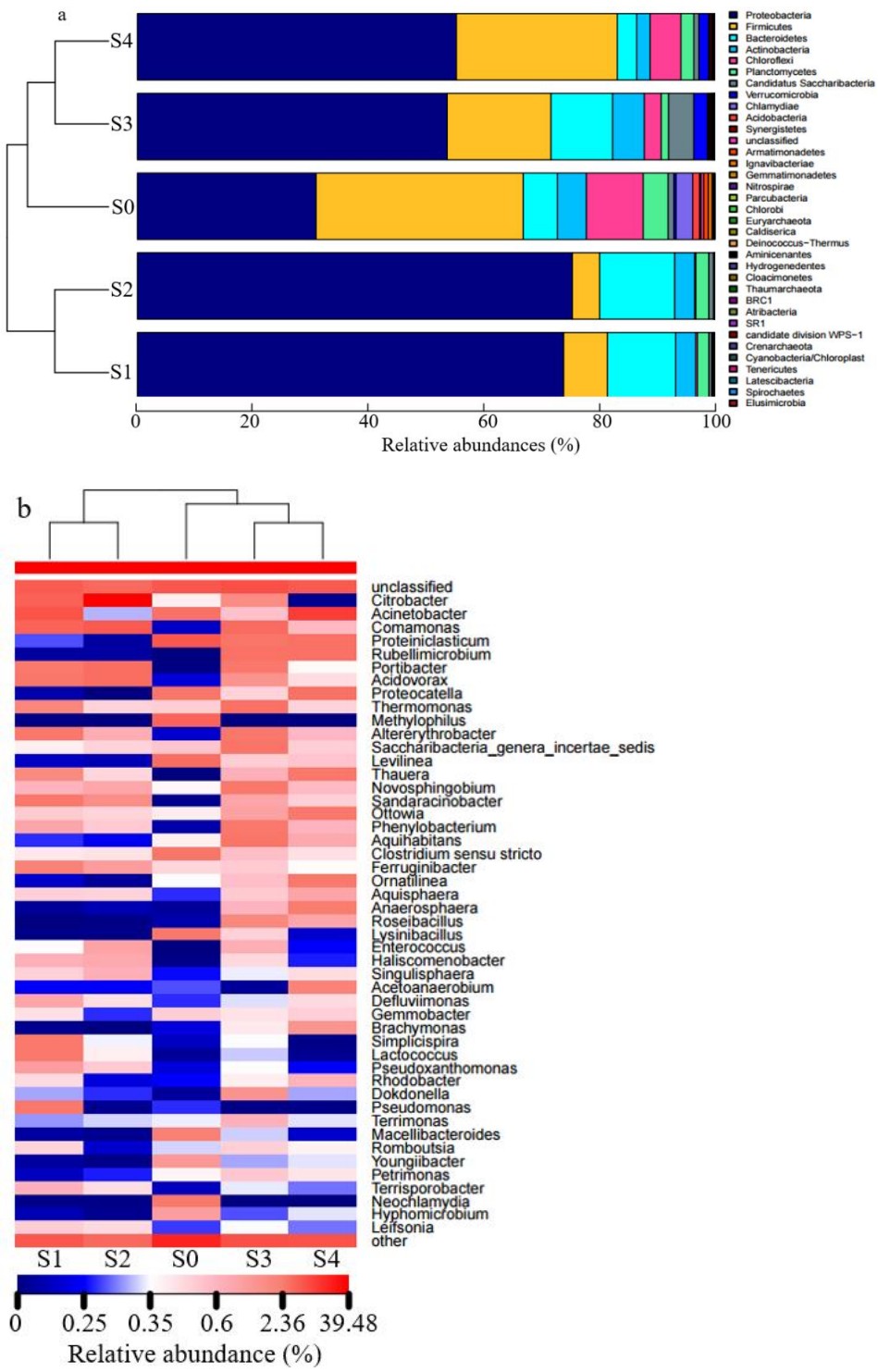

**Figure 5.** Microbial compositions in sludge samples at phylum (**a**) and genus levels (**b**).

*Acinetobacter* (Type 1683), the famous filamentous bacteria, was frequently reported in AS or AGS for filamentous bulking [9,45], totaling 4.82%, 18.82%, 3.25%, 4.05%, and 6.64% in sludge samples S0–S4, respectively (Figure 5b). This indicated that adding sludge micropowder could effectively restrain the filamentous extended growth of Type 1683 *Acinetobacter* (S1–S2) and low-level filamentous bacteria serve as a framework for enhancing the strength of aerobic granule and N removal [9]. This result was in contrast to previous work [10], which regarded Type 021N as the main extended filamentous bacteria in the AGS system using low ammonia nitrogen wastewater (20–30 mg/L). We hypothesized that wastewater composition and operation conditions played crucial roles in these two

filamentous proliferations. Additionally, we also detected Type 021N *Thiothrix* in the seeding sludge (S0) with a very low abundance of 0.27%, then did not find it in S1-S4 in this experiment, which further proved our hypothesis of high ammonia inhibiting microbial growth [5,46]. In addition, phylogenetic analysis demonstrated that all the samples S0–S4 could be divided into three groups (group I S0, group II S1 and S2, and group III S3 and S4) (Figure 5). Figure 6 shows the results of clustering analysis with principal co-ordinates analysis (PCoA) (Figure 6a) and non-metric multidimensional scaling (NDMS) (Figure 6b) based on OUT with unweighted UniFrac. All sludge samples were clustered in to three groups (G I-S0, G II-S1 and S2, and G III S3 and S4), which were identical to Figure 5, suggesting microbial composition was influenced by running conditions, wastewater characteristics, and sludge structure [42,46]. In short, *Citrobacter* and *Thauera* played a key role in the PND process for high-ammonia wastewater treatment in our work.

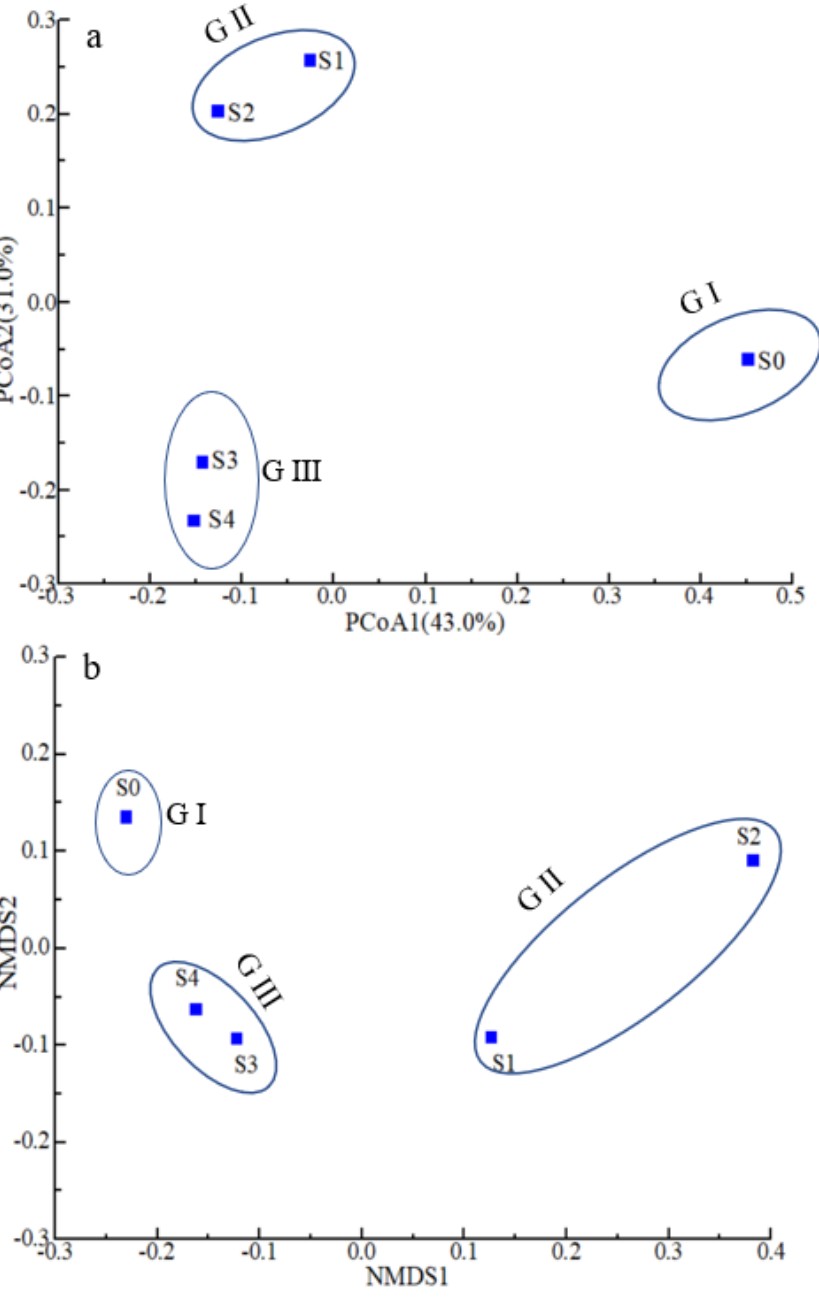

**Figure 6.** Plots generated from PCoA (a) and NDMS based on the OUT with unweighted UniFrac.

## 4. Conclusions

The bacterial composition and PND in a filamentous AGS-SBR using high-ammonia wastewater were comprehensively explored in this work. Type 1683 *Acinetobacter* with high percentage became the main filamentous bacteria in extended filamentous AGS, being effectively controlled by adding sludge micropowder and serving as skeleton for AGS recovery when kept at low levels. Meanwhile, the PND process was realized in an AGS-SBR treating high-ammonia wastewater by coupling aeration, stirring process, and sludge micropowder addition. Microbial analysis indicated *Citrobacter* and *Thauera* played an important role in the PND process for high-ammonia wastewater treatment, and microbic composition was markedly influenced by the running model, wastewater features, and sludge structure.

**Author Contributions:** Data curation, validation, and writing—original draft preparation, J.L. (Jun Liu), S.L., D.X. and W.H.; review and editing, Q.H. and W.C.; funding acquisition, resources, supervision, review and editing, J.L. (Jun Liu) and J.L. (Jun Li). All authors have read and agreed to the published version of the manuscript.

**Finding:** The authors would like to acknowledge the financial support from the State Key Laboratory of Pollution Control and Resource Reuse Foundation (PCRRF20007), Science and Technology Planning Project of Jiaxing (2021AY10080, 2021AD30166) and National Natural Science Foundation of China (51478433).

**Conflicts of Interest:** The authors declare no conflicts of interest. This study does not include experiments conducted on humans or animals.

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
