# Peer review of "Results of Adding Sludge Micropowder for Microbial Structure and Partial Nitrification and Denitrification in a Filamentous AGS-SBR Using High-Ammonia Wastewater"

_water, doi:10.3390/w15030508_

Round 1

Reviewer 1 Report

Is: 30. Du, R.; Peng, Y.Z.; Cao, S.B.; Li, B.K.; Wang, S.Y.; Niu, M. Mechanism and bacterial structure of partial denitrification with high nitrite accumulation. Appl. Microbiol. Biotechnol. 2016, 100; 2011-2021.

Should be:

40. Du, R.; Peng, Y.Z.; Cao, S.B.; Li, B.K.; Wang, S.Y.; Niu, M. Mechanism and bacterial structure of partial denitrification with high nitrite accumulation. Appl. Microbiol. Biotechnol. 2016, 100; 2011-2021.

It would be interesting to support the research result with chemical reactions.

Author Response

Dear reivewer,

The manscript was revised based on your suggestion.

Thanks and best wishes.

Reviewer 2 Report

In this paper, the author try to establish a method to think about the roles of sludge micropowder addition on microbial structure and partial nitrification & denitrification in an extended filamentous aerobic granular sludge-sequencing batch reactor with high ammonia wastewater. I think it is interesting for waste water treatment in the coming future. In the world, the waste water is becoming an intelligent work within population increasing and huge city growing. In my opinion, I have some doubts and comments as follows.

1. In the using of sludge micropowder, I think it is a carbon source for sludge treatment. So in this case, the author should support and discussion much in this part. Without much data support, I think this paper value is so limited for readers.

2. The correction samples are municipal sewage. But I still have a question that the too high ammonia. The author should support much evidence for this.

3. The experiment period is about half year. And the author separated four steps to discussion. I worry about the basis. It is an abnormal experiment and the real value is unclear.

        As a conclusion, the method for waste water treatment is nice and interesting. But I believe the authors have much data for this work. But in this paper, there are still some limitations for publish.

Author Response

Dear reviewer,

The manuscript was revised bassed on your suggestuons.

Thanks and best wishes.

Reviewer 3 Report

How to control excess growth of filamentous bacteria in AGS system has always been a research hotspot in the field of sewage treatment. In this manuscript, the authors utilize the sludge micro-powder to enhance the stability of AGS and also to realize the partial nitrification and denitrification. The results were very interesting. The manuscript is well organized and I think this study is important for the application of AGS in the real engineering. Before the publication, some parts of the manuscript should be improved:

1.        please give the detailed info about the addition of sludge micro-powder, 0.2 g/L is just the concentration, maybe should give the mass of the total sludge micro-powder.

2.        The nitrate in the influent and effluent was not detected, which was not sufficient for the conclusion about the PND, the authors should do the kinetic study for the one cycle.

In conclusion, I approve of its publication after major revision.

Author Response

Dear reviewer,

The mauscript was revised bassed on your suggestions.

Thanks and best wishes.

Round 2

Reviewer 2 Report

After correction, I think it is better than before. And it is interesting for readers.